# Exploring Older Adults’ Perceived Affordability and Accessibility of the Healthcare System: Empirical Evidence from the Chinese Social Survey 2021

**DOI:** 10.3390/healthcare11131818

**Published:** 2023-06-21

**Authors:** Lu Chen, Miaoting Cheng

**Affiliations:** 1School of Journalism and Communication, Guangzhou University, Guangzhou 510006, China; sjccl@gzhu.edu.cn; 2Department of Educational Technology, Faculty of Education, Shenzhen University, Shenzhen 518060, China

**Keywords:** perceived affordability and accessibility, older adults, public perception

## Abstract

The difficulties in accessibility and affordability of healthcare services have posed persistent challenges to the Chinese government ever since the 1990s. This research aimed to explore the perceived accessibility and affordability of the healthcare system, which is also referred to as the social problem of *Kan Bing Nan*, *Kan Bing Gui* among older Chinese adults. The research, based on 2169 sample data points from the Chinese Social Survey (CSS) 2021 database, explored the objective and subjective factors in constructing the public perceptions of accessibility and affordability of the healthcare system among both older adult users and older adult non-users of healthcare services, including healthcare expenditure variables, patient experience variables, financial protection variables, and social evaluation variables. The study utilized binary logistic regression analysis to investigate how four different sets of independent variables impact the perceived affordability and accessibility of the healthcare system by older adults. The research found that *Kan Bing Nan*, *Kan Bing Gui* was considered the most pressing issue among the older adults in China. Self-reported medical burdens, the cost of healthcare services, and the perceived fairness of public healthcare services were significantly associated with older adults’ perceived affordability and accessibility of the healthcare system.

## 1. Introduction

The healthcare system in China has undergone dramatic reform since the establishment of the People’s Republic of China in 1949. The most recent round of healthcare reform started in the year 2009, aiming to provide universal coverage of basic healthcare by the end of 2020 and emphasizing government-led, people-centered healthcare [1]. Although China has made great achievements in increasing healthcare accessibility, improving financial protection, and reducing health inequity since 2009, satisfaction of the health system did not increase from 2010 to 2016 [2]. The increasing burden associated with an ageing population leads to gaps in fulfilling the needs of healthcare services [3]. According to the seventh national census in 2020, the population aged 60 and older accounted for 18.70% of the population in China [4]. China is moving toward a superannuated society, bringing about a heavy disease burden and increasing the need for care among older people with chronic diseases and disabilities [5]. As the proportion of the older population dramatically expands, difficulties in the accessibility and affordability of healthcare services exist in terms of high health expenses, low benefits of health insurance, and a shortage of healthcare services [6].

The difficulties in the accessibility and affordability of healthcare services have posed persistent challenges to the Chinese government ever since the 1990s [7]. In China, the discontent of the public towards the healthcare system is well-known as *Kan Bing Nan*, *Kan Bing Gui* (“看病难, 看病贵”), which means an “insurmountable access barrier to health care, insurmountably high health costs” [8]. The saying has been adopted in the government policy documents, and solving the difficulties in accessibility and affordability has been treated as the main goal of healthcare reform [9]. However, as the major policy outcomes of healthcare reform, little is known about the public perception of the accessibility and affordability of the healthcare system after years of reform, especially among vulnerable groups such as older adults. Therefore, this study aims to explore the following questions: First, how do older adults perceive the accessibility and affordability of the healthcare system in China? Second, what are the factors that are associated with their perceptions of accessibility and affordability?

## 2. Literature Review

### 2.1. Healthcare System Reform in China

The social problem of *Kan Bing Nan*, *Kan Bing Gui* caused widespread social discontent with healthcare and gained serious political attention at the end of the 1990s [7,8]. In response to the public discontent, after 2009, China’s efforts to build up Universal Health Coverage (UHC) emphasized pursuing better equity, quality, and affordability [10]. To improve access and quality while lowering the cost of healthcare, the new round of healthcare reform in 2009 focused on three primary areas, which are health insurance, drug pricing, and public hospitals [9]. Many policies have been implemented to expand insurance coverage and improve the efficiency of healthcare delivery by establishing primary healthcare (PHC) facilities [3].

First, to solve the problem of affordability, the government endeavored to expand the coverage of healthcare insurance. Three insurance schemes, i.e., the New Rural Cooperative Medical Scheme (NRCMS) in rural areas and the Urban Employee Basic Medical Insurance (UEBMI) and Urban Resident Basic Medical Insurance (URBMI) in urban areas, together constituted China’s basic social medical insurance system [11]. These schemes are divided into individual accounts and pooled funds. The medical expense incurred in a settlement period (generally 1 year) will be calculated by stages and paid cumulatively according to the hospital grade (community, primary, secondary, tertiary). Then, the pooled fund and individuals share the expenses following the prescribed proportion [10]. Second, to solve the problem of accessibility, the government also increased investment in primary healthcare facilities to promote equal access and universal healthcare coverage [12]. As a major target of health reform in 2009 and a core component of the government’s “Healthy China 2030” strategy [1], primary care facilities are expected to alleviate the over-utilization of secondary and tertiary care by tackling frequently encountered medical conditions at community levels [13]. The reform has had significant achievements in increasing health insurance coverage, lowering out-of-pocket expenditure, and providing essential public health services to all citizens through primary healthcare facilities, which make the healthcare services be more affordable and accessible [14].

However, many challenges still exist in the affordability and accessibility of healthcare services in China. Generally, the current healthcare system seems not to favor vulnerable groups such as older adults. First, benefits and reimbursement rates vary across different health insurance schemes, which suggests disparities in benefits between rural and urban residents, different socio-economic groups, formal sector employees and others, and disparities due to age differences of insured individuals in each pool [8]. Rural and unemployed residents receive fewer benefits from healthcare insurance schemes because NRCMS for rural residents and URBMI for urban unemployed (including the elderly and children) have a lower actual reimbursement rate than that of UEBMI for urban employees (and retired employees). According to an existing study, the estimated actual reimbursement rate for urban employees exceeded 63% while the estimated actual reimbursement for urban residents and rural residents was about 37% in the year 2016 [10]. Second, people still prefer to seek medical services in hospitals rather than primary care institutions, which leads to overcrowding in larger hospitals and rapidly rising service costs [15]. The underutilization of primary care facilities is very common countrywide [16].

Older adults are susceptible to the impacts brought about by these disparities in health insurance benefits and tiered healthcare service utilization. Generally, the reimbursement rate of primary health institutions in many provinces is higher than that of tertiary institutions, and local government intentionally tries to encourage patients to utilize primary health institutions through such medical insurance policies [17]. However, elderly patients are more likely to have diseases requiring comprehensive checkups, treatments, and medicines provided by large hospitals, which may not be listed as reimbursable items in the healthcare insurance scheme they enrolled in and therefore receive lower actual reimbursement rates [18]. Due to declining physical and mental conditions, they have an increasing demand for inpatient and other services for disabilities/chronic diseases in tertiary hospitals, which make them prone to catastrophic health expenses [19]. Health care for the elderly is under pressure due to the disparities of different healthcare insurance schemes, insufficient resources available for elderly care, as well as the demand for long-term medication and outpatient coverage [20,21]. According to these existing studies, it seems that older adults are likely to encounter difficulties in the accessibility and affordability of healthcare services due to their vulnerability during the aging process.

### 2.2. Public Perceptions of the Healthcare System

The social problem of *Kan Bing Nan*, *Kan Bing Gui* is not only about actual difficulties in accessibility and affordability faced by the public, but it also represents public perceptions constructed by academic literature and policy proposals vacillating between the state or market debates [22]. Therefore, in addition to the objective evaluations of accessibility and affordability, it is also important to examine the perceived accessibility and affordability as a dimension of the public perceptions of the healthcare system.

One of the prolific research areas exploring public perceptions of the healthcare system is public satisfaction research. Public satisfaction is a general subjective evaluation of the healthcare system for all citizens, including users and non-users of healthcare services [23]. Existing studies found that objective measures, such as access and expenditure, as well as the subjective expectation of these objective measures constitute the sub-dimensions of satisfaction with the healthcare system [24]. In other words, satisfaction with the healthcare system may depend more on specific cultural and political factors external to the actual performance of the system itself [25]. For example, a study on public opinion in 61 countries found that peoples’ perceptions of accessibility and affordability were significant dimensions of overall public opinions towards the healthcare system [26]. Citizens’ expectations and political debates may influence the public satisfaction with the overall healthcare system or with specific domains such as affordability and accessibility [27].

These cross-national studies revealed the influence of subjective factors on the formation of the public perceptions of the affordability and accessibility of the healthcare system. Some empirical studies in the Chinese context demonstrated the role of subjective factors, such as the beliefs of the healthcare system, in constructing the public perception of the healthcare system. In China, satisfaction with the healthcare system is strongly associated with the subjective belief in personal responsibility for meeting healthcare costs while negatively associated with perceptions of unequal access as well as unethical service providers [28]. Mass media in China have long been reporting negative coverage and dramatizing unethical misconducts of hospitals and health professionals, which create negative public opinions towards healthcare services [29,30,31]. Generally, both cross-national studies and existing empirical studies in the Chinese context revealed that public perceptions were socially and culturally constructed beyond the actual performances of the healthcare system.

### 2.3. Patient Experience and Public Perception

Patients are users of healthcare services, and thus patient experience is associated with the public perception of the healthcare system. For example, in a study of 21 European countries, those who had used healthcare services in the last 5 years were more satisfied than those who had not [32]. Patient experience can be conceptualized as both patients’ experiences of care and as feedback from patients about those experiences [33]. There are many debates about the extent to which satisfaction with healthcare services is explained by patient experience. Some studies have concluded that much of the variation in the satisfaction with health services is explained by patient experience, while other studies posit that patient experience accounts for only a small fraction of the variation in health service satisfaction [34]. Patient experience can be measured by various items, such as waiting time, doctor–patient communication, and staff responsiveness [33].

Current studies on patient experience in China mainly explore the doctor–patient relationship and communication. In China, public hospitals are not fully sponsored by the government to pursue public interests or privately owned profit-maximizing entities. As a result, sometimes, the healthcare providers tend to pursue their own interests through overtreatment, overprescription, and getting red-envelope money (i.e., a very big tip intended to ensure good service) from patients [35]. These misconducts caused difficulties in the accessibility and affordability of healthcare services for patients while bringing about negative patient experiences. For instance, a prevalently and commonly observed phenomenon is called ‘*guanxi* jiuyi’ (medical *guanxi*), which involves patients leveraging their ‘*guanxi*’ (personal connections) to establish mutual relationships with physicians when seeking healthcare. Patients and their families believe this facilitates access to experienced medical specialists and reduces the financial and medical risks of overdiagnosis and overtreatment [36]. Such distorted doctor–patient relationships usually lead to a perception of unfairness and injustice in the medical care system [37]. However, current studies mainly focus on trust issues in doctor–patient relationships while seldom discussing other aspects of patient experience, such as patients’ feedback on waiting time and service quality. Moreover, little is known about the relationship of patient experience at the individual level to overall public perception or a specific dimension of public perception, such as perceived accessibility and affordability at the macro level.

### 2.4. Conceptual Framework

In sum, our research aimed to explore the perceived accessibility and affordability of the healthcare system, which also refers to the social problem of *Kan Bing Nan*, *Kan Bing Gui* among older Chinese adults. Based on existing studies, we explored the objective and subjective factors in constructing the public perceptions of the accessibility and affordability of the healthcare system among both elder users and elder non-users of healthcare services, including healthcare expenditure variables, patient experience variables, financial protection variables, and social evaluation variables. Objective factors refer to the factual data, including out-of-pocket medical expenditure in the last year and healthcare insurance schemes enrolled in by the respondents. Subjective factors are self-reported and evaluative data, including the self-reported medical burden, patient experience, and social evaluation variables.

As we have reviewed the existing literature above, we generalize some main factors that may influence the perceived accessibility and affordability from existing studies to construct our conceptual framework in Figure 1. We categorize the factors into four sets of variables, which include both subjective and objective factors and correspond to our literature review from Section 2.1, Section 2.2 and Section 2.3.

## 3. Method

### 3.1. Data and Sampling

The data for this study were obtained from the Chinese Social Survey (CSS) open database. The CSS is a comprehensive cross-sectional survey project conducted by the Institute of Sociology at the Chinese Academy of Social Sciences and provides nationally representative data. The CSS survey is a biannual longitudinal survey to collect labor and employment data, family and social life information, social attitudes, and other aspects. A multi-stage stratified probability sampling method was used to select and interview a representative sample that reflected the profile of households across China. To ensure data quality and a high response rate, a research team from universities and scientific research centers was established, who provided 3–5 days of training to the data collectors and developed the quality control system. Written informed consent was obtained from all respondents, and the study objectives were also clearly introduced. The study utilized the unaltered dataset from the CSS 2021, encompassing 100,136 individuals residing in 584 villages/communities and 152 counties/districts across 64 provinces/cities/autonomous regions in China. Given the focus on older Chinese adults, the selection criteria were limited to respondents aged 60 years or older (born in or after 1961). Subsequently, the variables were carefully screened, matched, and processed, with the respondents’ birth year recorded in the questionnaire, resulting in a final sample size of 2169 data points.

### 3.2. Variables

#### 3.2.1. Dependent Measure

In this study, the perceived affordability and accessibility of the healthcare system was the dependent variable. Respondents were asked to answer the question: “What do you think is the most pressing social problem in our country?” The dependent variable was constructed as a binary variable and was assigned a value of 1 if the older adults selected *Kan Bing Nan*, *Kan Bing Gui* as China’s most pressing social problem and a value of 0 if not.

#### 3.2.2. Independent Measures

This study identified four sets of independent variables. The first set is healthcare expenditure variables, including medical expenditures and self-reported medical burdens. Medical expenditure is the objective indicator of household out-of-pocket medical expenditure. The medical expenditure variable was developed by asking respondents to fill in the total amount their family spent on healthcare in the last year (2020) after insurance reimbursement and therefore was constructed as a continuous variable. The self-reported medical burden is the subjective indicator of unaffordable payment. The self-reported medical burden variable was developed by asking respondents to report whether in the last twelve months they or their family has had difficulties in paying for healthcare services and feel the payment is unaffordable. This was constructed as a binary variable with two categories (No = 0, Yes = 1).

The second set is patient experience variables, including difficulties in distance, waiting time, cost, and quality, which were developed by asking respondents to report to what extent they encountered difficulties the last time when seeking healthcare services in terms of distance, waiting time, cost, and quality and which were constructed as a 4-point Likert scale variable ranging from 1 (“very serious”) to 4 (“did not encounter this difficulty”).

The third set is financial protection variables, including insurance coverage and insurance type. The insurance coverage variable was developed by asking respondents to report whether they had enrolled in any medical insurance schemes provided by the government; this was constructed as a binary variable with two categories: 1 (Yes) and 0 (No). For the insurance type variables, the older adults were asked to report whether they had enrolled in the following types of insurance schemes: Urban Employee Basic Medical Insurance (UEBMI) (No = 0, Yes = 1), Urban Resident Basic Medical Insurance (URBMI) (No = 0, Yes = 1), Government Insurance Scheme (insurance scheme for civil servants, GIS) (No = 0, Yes = 1), New Rural Cooperative Medical Scheme (NRCMS) (No = 0, Yes = 1), and critical illness insurance for urban and rural residents (CICURR) (No = 0, Yes = 1).

The fourth set comprises social evaluation variables, including satisfaction with healthcare insurance, satisfaction with local healthcare service delivery, and the perceived fairness of public healthcare services. The satisfaction with healthcare insurance variable was developed by asking respondents to report to what extent they were satisfied with the healthcare insurance provided by the government, which was constructed as a 10-point Likert scale variable ranging from 1 (“very unsatisfied”) to 10 (“very satisfied”). The satisfaction with local healthcare service delivery variable was developed by asking respondents to report to what extent they were satisfied with the local government’s performance in healthcare service delivery, which was constructed as a 4-point Likert scale variable ranging from 1 (“very unsatisfied”) to 4 (“very satisfied”). The perceived fairness of healthcare services variable was developed by asking respondents to report to what extent they perceived the fairness of the public healthcare services, which was constructed as a 4-point Likert scale variable ranging from 1 (“very unfair”) to 4 (“very fair”).

#### 3.2.3. Statistical Analysis

First, a descriptive analysis was employed to gain an overall profile of the measurement variables and the older adults. Subsequently, the point-biserial correlation was utilized to investigate the connections between social evaluation and the affordability and accessibility perceived by older adults. Furthermore, considering that the dependent variable, perceived affordability and accessibility of the healthcare system, was presented as a binary variable, binary logistic regression analysis was conducted to examine how the four sets of intended independent variables impacted older adults’ perceived affordability and accessibility of the healthcare system. All data analyses were carried out using IBM SPSS 25.

## 4. Findings

### 4.1. Descriptive Characteristics of the Sample

The descriptive statistics of the variables of interest in this study are presented in Table 1. Finally, this study obtained a sample comprising 1042 male and 1127 female older adults, with an average age of 64.92 years (SD = 2.745). The descriptive results showed that 54.5% of the older adults perceived *Kan Bing Nan*, *Kan Bing Gui*. We further conducted a descriptive analysis of China’s fourteen pressing social problems. The fourteen pressing social problems included unemployment, *Kan Bing Nan*, *Kan Bing Gui*, old-age security, education expenses, inequality between rich and poor, rising prices, expensive housing prices, public security, decreasing social trust, corruption, environmental pollution, the safety of food and drugs, injustice in land expropriation, and unequal treatment of rural–urban immigrant workers. It should be noted that *Kan Bing Nan*, *Kan Bing Gui* has become the most pressing issue among the older adults in China, as more than half of the older adults selected this as the most pressing societal issue in China (54.5%), followed by rising prices (25.9%) and inequality between the rich and poor (25.5%), calling for our attention to improve older adults’ affordability and accessibility to healthcare services.

### 4.2. Logistic Regression Model of Perceived Affordability and Accessibility

Point-Biserial correlation analyses were conducted to investigate the association between the social evaluation variables of the older adults and their perception of affordability and accessibility. Given that perceived medical expenditure violated the assumption of normality, it was not included in the correlation analysis. The results showed that the older adults’ perception of affordability and accessibility was significantly and negatively related to all of the social evaluation variables, including perceived difficulties in distance (r = −0.119, *p* < 0.001), perceived difficulties in waiting time (r = −0.150, *p* < 0.001), perceived difficulties in cost (r = −0.301, *p* < 0.001), perceived difficulties in quality (r = −0.136, *p* < 0.001), satisfaction with healthcare insurance (r = −0.180, *p* < 0.001), satisfaction with local healthcare service delivery (r = −0.146, *p* < 0.001), and perceived fairness of healthcare services (r = −0.201, *p* < 0.001). The results indicated that the more older adults perceived *Kan Bing Nan*, *Kan Bing Gui*, the more likely they perceived difficulties in terms of distance, waiting time, cost, and quality in seeking healthcare services, the less likely they were satisfied with healthcare insurance and local healthcare service delivery, and the less likely they perceived fairness of public healthcare services.

To further examine the factors influencing the older adults’ perceived affordability and accessibility of the healthcare system, binary logistic regression analysis was utilized to examine the effects of the intended independent variables on perceived affordability and accessibility while controlling for demographic characteristics. In the base model (Model 1), demographic variables, including gender, age, education level, total annual income, and residency, were identified as common confounding factors and were examined. To investigate the impact of the four sets of independent variables, namely, healthcare expenditure variables (Model 2), patient experience variables (Model 3), social evaluation variables (Model 4), and financial protection variables (Model 5), separate models were constructed. The results of the Hosmer and Lemeshow chi-square test indicated that all models, i.e., Model 1 (χ^2^ (8) = 12.42, *p* = 0.13), Model 2 (χ^2^ (8) = 7.90, *p* = 0.44), Model 3 (χ^2^ (8) = 4.75, *p* = 0.78), Model 4 (χ^2^ (8) = 16.26, *p* = 0.04), and Model 5 (χ^2^ (8) = 2.48, *p* = 0.96), did not demonstrate statistical significance, indicating that all models had a good fit. Moreover, the results of the likelihood ratio chi-square test showed that Model 1 (χ^2^(6) = 18.02, *p* < 0.001), Model 2 (χ^2^ (8) = 88.62, *p* < 0.001), Model 3 (χ^2^ (12) = 157.11, *p* < 0.001), Model 4 (χ^2^(17) = 117.74, *p* < 0.001), and Model 5 (χ^2^(20) = 129.01, *p* < 0.001) had a significant improvement in fit over the intercept-only null model. The descriptive results of the binary logistic regression models were also summarized and are presented in Table 2.

The demographic variables were included as common confounding variables in Model 1; the results showed that all of the demographic characteristics did not significantly affect the older adults’ perception of affordability and accessibility except for the education level in Model 1 and Model 2. Moreover, the Nagelkerke R^2^ of Model 1 was only 0.01, and the explanatory power increased from 0.01 to 0.22 in Model 5, indicating that the inclusion of the four sets of intended independent variables better explained the dependent variable. For Model 2, only healthcare expenditure variables were included. The results showed that medical expenditure (B < 0.01, OR = 1.00, *p* > 0.05) did not significantly affect the older adults’ perception of affordability and accessibility, whereas the older adults’ self-reported medical burden significantly influenced the dependent variable. More specifically, with the use of no medical burden as the reference group (B = 0.82, OR = 2.28, *p* < 0.001), for every one unit increased in the older adults’ self-reported medical burden, the odds ratio of increasing the older adults’ perception of affordability and accessibility by one additional level increased by 128%.

For Model 3, the healthcare experience variable was also included in the model. In addition to the medical expenditure variable, the older adults’ encountered difficulties in seeking healthcare services in terms of distance (B = −0.01, OR = 0.99, *p* > 0.05), waiting time (B = −0.05, OR = 0.95, *p* > 0.05), and quality (B = −0.02, OR = 1.02, *p* > 0.05) did not significantly affect the dependent variable. Nevertheless, the self-reported medical burden (B = 0.48, OR = 1.61, *p* < 0.001) was a significant and positive predictor of the dependent variable. In addition, the cost of healthcare services (B = −0.52, OR = 0.60, *p* < 0.001) was a negative and significant predictor of the dependent variable. When the cost of healthcare services as a very serious difficulty was used as the reference group, for every one unit increase in the older adults’ difficulty in cost, the odds ratio of increasing the older adults’ perception of affordability and accessibility decreased by 42%. In other words, if the older adults did not encounter any difficulty in the cost of healthcare services, they were not likely to perceive *Kan Bing Nan*, *Kan Bing Gui* as the most pressing social problem in China.

For Model 4, we included the financial protection variables in terms of the enrollment of the five insurance types, and the results revealed no significant effects of the financial protection variables on the dependent variable. Moreover, whereas the Nagelkerke R^2^ increased from 0.06 to 0.15 from Model 2 to Model 3, the R^2^ had only a very limited improvement from Model 3 (0.15) to Model 4 (0.17). Finally, the social evaluation variables were included in Model 5. Similarly, the results also supported the reported medical burden as a significant and positive predictor (B = 0.24, OR = 1.27, *p* < 0.05) and encountered difficulty in the cost of healthcare services as a significant and negative predictor (B = 0.60, OR = 0.55, *p* < 0.001), whereas none of the financial protection variables had a significant effect on the dependent variable. As for the social evaluation variables, satisfaction with healthcare insurance (B = −0.04, OR = 0.96, *p* > 0.05) and satisfaction with local healthcare service delivery (B = −0.11, OR = 0.89, *p* > 0.05) did not have a significant association with the dependent variable. In contrast, the older adults’ perceived fairness of public healthcare services was strongly associated with the older adults’ perceived affordability and accessibility (B = −0.37, OR = 0.69, *p* < 0.01). More specifically, for every one unit increase in older adults’ perceived fairness of public healthcare services, there was a 31% decrease in the odds ratio of increasing their perception of affordability and accessibility, indicating that the more the older adults perceived healthcare services as fair, the less likely they perceived *Kan Bing Nan*, *Kan Bing Gui*.

## 5. Discussion

Our research finds that 54.5% of the older adults considered *Kan Bing Nan*, *Kan Bing Gui* as the most pressing social problem in China, which is far more than those who selected other social problems listed in the CSS 2021 questionnaire. This implies that due to the declining health conditions in the later stage of life, older adults tend to be more concerned about the social problems related to healthcare services rather than other social issues related to earlier stage of life, such as housing, employment, and education.

It should be noted that our research on perceived accessibility and affordability includes both users’ and non-users’ perceptions of the healthcare system. Some respondents had not used healthcare services in the past year and incurred no payments. The variable of medical expenditure is more likely to be influenced by the extreme value of the payment amount. Thus, we adopted medical expenditures as objective indicators and the self-reported medical burden as a subjective indicator of healthcare expenditures. Our research found that only the self-reported medical burden significantly impacted the older adults’ perceived affordability and accessibility of the healthcare system, while actual household medical expenditures in the last year did not. The results indicate that those older adults who perceived themselves as having generated unaffordable expenditure last year were likely to perceive inaccessibility and unaffordability in the healthcare system. Such results also implied that the perceived affordability and accessibility of the healthcare system were closely related to the users who utilized the healthcare services. However, it is far from certain to conclude that objective medical expenditure in the past year did not matter to perceived accessibility and affordability. There are many objective measures of patients’ financial risks and medical burdens. For example, the World Health Organization defined catastrophic health expenditure as an out-of-pocket health payment for health care equaling or exceeding 40% of a household’s capacity to pay [19]. The complexity lies in the social context that China has very high individual savings rates, and the Chinese elderly tend to accumulate savings throughout their whole life cycle to deal with the reduced income and increasing medical expenditures in their later life [38]. Some existing studies that adopted objective measures such as healthcare expenditure/disposable income per capita to measure the economic burden of healthcare also found that it was not the major driving factor for healthcare satisfaction [39]. We argue that it is possible because the precautionary savings of a family reduce medical expenses and alleviate the dissatisfaction with affordability. An existing study using the health-cost-to income ratio as the objective measure and perceived financial difficulty as the subjective measure of the financial burden of lung cancer patients found that the proportion of patients with financial burdens was different when measured using objective or subjective means, which indicated that it could be helpful to capture the actual financial burden by examining both dimensions [40]. We suggest that it is important to pay attention to the subjective evaluation of affordability among older Chinese adults. If the respondents perceived unaffordable medical burdens, it could be an alarming signal that the family’s savings have run out for healthcare services, and catastrophic healthcare expenditure may occur. In our study, a fairly large proportion of the older adults reported that they perceived medical burdens (41.4%), which were significantly related to the perceived affordability and accessibility of the healthcare system.

Our research results further demonstrated the relationship between patient experience and perceived accessibility and affordability. According to a study on patients’ assessment of healthcare services in the Chinese context that categorized 34 indicators into strong, medium, weak, and undetermined indicators, the researchers found that due to the shortage of medical resources in China, patients paid little attention to indicators, such as waiting time, the crowdedness of the hospital, and the convenience of transportation. Rather, they were more concerned about the convenience of paying, the transparency of the fees, and the rationality of the medical fee [41]. Our results also found that, compared to the cost of healthcare services, other indicators of patient experience, such as distance, waiting time, and quality, did not have a significant relationship with perceived accessibility and affordability. Thus, although the saying *Kan Bing Nan*, *Kan Bing Gui*, which means “insurmountable access barrier to health care, insurmountably high health costs [8]” and is usually mentioned in parallel in the same sentences in government documents, it is the financial barrier that makes the older adults feel difficulty in gaining access to and paying for healthcare services. In other words, the difficulty in utilizing healthcare services lies in the difficulty in affordability. Our research results also resonated with existing studies on the public perception of the healthcare system in China based on the CSS database, which showed that the perceived affordability of the last medical visit was strongly associated with a patient’s perception of the health system, while accessibility measures (e.g., waiting time and distance) were less likely to influence the public perceptions of health care [42]. Our results show that relieving patients’ financial burdens is still one of the key challenges for the successful implementation of healthcare reform in China.

Our results revealed no significant effects of the financial protection variables on the older adults’ perceived accessibility and affordability of the healthcare system. According to existing studies, segmentation by urban–rural and employment status relates to different benefits of healthcare insurance packages in China [43]. Because of the fragmentation in the healthcare insurance system, the reimbursement levels and the benefit packages have disparities, which drove the government to launch pilots for the integration reforms of healthcare insurance in several locations in 2007 and deepened the reform in 2016 [44]. Fragmentation in social health insurance schemes is an important factor for inequitable access to health care and financial protection for people covered by different health insurance schemes [45]. Some existing studies showed that enrollment in healthcare insurance significantly impacted the public perception of the healthcare system [28]. However, the results of our study resonated with other studies that did not find a significant relationship between enrollment in healthcare insurance schemes and the public perception of the healthcare system. For example, some studies found that although the population preferred a lower out-of-pocket expense, the ratio of out-of-pocket expenses had no significant impact on public satisfaction with the healthcare system [23]. Medical insurance enrollment was not a significant factor in evaluating the accessibility of medical resources [46]. The possible explanation could be that due to the vulnerability of older adults, the financial protection of these healthcare insurance schemes for the older adults was still insufficient. Existing research showed that, due to the effect of cream-skimming, the biggest gap lies in hospital access, and the payment of China’s healthcare service indicates disparities between senior cadres and the general public [47]. Our results indicated that older adults experienced comparable difficulties in accessibility and affordability irrespective of the type of healthcare insurance scheme in which they enrolled. It is also possible for the cream-skimming effect that satisfaction with healthcare insurance was not significantly associated with perceived accessibility and affordability.

Interestingly, satisfaction with local healthcare service delivery was not significantly associated with the older adults’ perceived accessibility and affordability of the healthcare system, while the older adults’ perceived fairness of public healthcare services was strongly associated. In China, the distribution of healthcare resources has regional disparities between urban and rural and developed and less developed provinces. Since the government expenditure on healthcare at sub-national levels accounts for ~90% of the total government expenditure on healthcare, inequity in public expenditure exists at sub-national levels [48]. Due to health finance decentralization and uneven economic development in China, local governments in less developed regions cannot fulfill their responsible investments in the healthcare sector [49]. However, our results implied that compared to satisfaction with healthcare services within immediate reach at the local level, the older adults placed more weight on the perceived fairness of the overall healthcare services. Our results are in line with the existing studies, indicating that perceived social equality is a promoting factor in the accessibility evaluation of medical resource allocation [46]. Different from equality or equity, which refers to the gap in distribution, fairness in Chinese usually includes the meaning of distributive justice, which implies that one gets what one deserves based on a perspective of self-interest [50]. Fairness in Chinese is inclined toward egalitarianism, which is a feature of privilege deprivation rather than redistribution [51]. Egalitarianism has been ingrained in the traditional Chinese cultural context. In China, the ancient saying *Bu Huan Gua Er Huan Bu Jun* (“不患寡而患不均”) means “no worry about scarcity but unevenness” [52]. Although perceived fairness in healthcare increased from 2006 to 2009 [42], many problems with allocating and distributing healthcare resources still exist. For example, profit-orientated practices widely exist in public institutions, and *guanxi* is more often used to access hospital-based services, which could be a potential factor contributing to the inequality of health resource allocation [53]. In addition to significant inequality in the geographic distribution of health resources, the rich are more likely to use well-resourced hospitals for outpatient care, while poorer people are more likely to use poorly resourced primary care institutions for inpatient care [54]. Due to the limited effect of health insurance on reducing the inequality in healthcare utilization, the pro-rich horizontal inequities in both the probability and frequency of health services among middle-aged and older adults exist [55]. Such uneven distribution and allocation of healthcare resources may generate the perception of unfairness and injustice.

## 6. Conclusions

There are many research gaps in examining the current public perception of the healthcare system in China, especially accessibility and affordability, which has long been considered the major goal and outcome of healthcare reform by policymakers. To fill the gap, our research investigated the perceived accessibility and affordability of the healthcare system among older adults and revealed the key determinants of these perceptions. Based on 2169 sample data points from the CSS 2021 survey, the results of descriptive analyses revealed that 54.5% of the older adults considered *Kan Bing Nan*, *Kan Bing Gui* as the most pressing social problem in China; the results of binary logistic regression analyses revealed that self-reported medical burdens, perceived difficulties in cost, and perceived fairness of healthcare services influenced the older adults’ perception of affordability and accessibility.

The results reflected that although healthcare reform in China has achieved great progress in the past years, for vulnerable groups such as older adults, perceived difficulties in affordability continue to be the main barrier for them to access healthcare services. Our research showed that enrollment in different healthcare insurance schemes did not significantly impact perceived affordability and accessibility. To provide better financial protection, expanding the coverage of healthcare insurance, raising the reimbursement rate of healthcare services, and reducing the out-of-pocket burden are reasonable measures to solve the affordability problem. However, some studies found that the current healthcare insurance schemes contributed to increasing pro-rich inequity in health service utilization [55]. The healthcare insurance schemes did not significantly reduce the out-of-pocket payments, and the schemes with more benefits even tended to induce higher health costs [56,57]. Thus, we argue that in addition to increasing coverage and the reimbursement rate, effective measures should be taken to reduce the pro-rich inequity and cream-skimming effects to ensure the most vulnerable and disadvantaged older adults get better financial protection. Increasing the affordability of healthcare services among vulnerable groups may be a main challenge for healthcare reform in China.

Our research also illustrated that it is necessary to consider the subjective evaluation of healthcare expenditures such as the self-reported medical burden for a better understanding of the actual financial burden among the older adults in China. Due to the high individual saving rates in China, the function of precautionary savings in buffering the medical expenditure burden should be addressed. Subjective evaluation, such as self-reported medical burden, could be a signal of running out of lifetime savings for healthcare services. Thus, we further call for more nuanced studies on the mediating role of household precautionary savings in the relationship between medical expenditure and household consumption to provide more effective financial protection. Our research also indicates that subjective beliefs may play a significant role in constructing the public perception of the healthcare system. We found that the perceived fairness of public healthcare services was strongly associated with perceived accessibility and affordability among the older adults. Therefore, we argue that to alleviate the perceived unfairness, it is important for policymakers to balance the allocation of healthcare resources and establish a unified healthcare insurance scheme reducing disparities in different funding pools. It is also important to eliminate the privilege and misconduct in access to healthcare services such as medical *guanxi*.

Our research has some limitations. First, in the CSS 2021 database, a lack of details about the actual reimbursement rate of healthcare service utilization in the last year makes it difficult to evaluate the effectiveness of financial protection of each healthcare insurance scheme. In China, the actual reimbursement rate of healthcare services not only depends on enrollment in different schemes but also depends on various factors such as inpatient or outpatient service utilization, medicine prescribed, treatment received, level of hospital, and local or non-local service spots. The actual reimbursement rate may cause the disparities of benefits and financial protection among different medical insurance schemes. Therefore, it is necessary for more databases to collect these data for future research. Especially, since 2016, the New Rural Cooperative Medical Scheme (NRCMS) and Urban Resident Basic Medical Insurance (URBMI) began to merge into the new Urban and Rural Resident Basic Medical Insurance scheme [58]. It is important to explore the change of the actual reimbursement rate under the new scheme to evaluate the performance of a unified medical insurance system. Second, the CSS 2021 database collected data in 2020 and 2021. The outbreak of the COVID-19 pandemic may have restricted healthcare service utilization among older adults and thus reduced healthcare expenditures in the past year. Further, the public perception of the healthcare system is more likely to change during the pandemic. However, due to a lack of specific details about healthcare-seeking behavior and public perception during the pandemic, we did not study the impacts of COVID-19 on the perceived accessibility and affordability of the healthcare system. Third, this study used the open-source CSS 2021 dataset, which mostly measures the identified variables with observed items. A future study could use mixed methods to develop a more comprehensive measurement scale to assess the older adults’ experience and perceptions of healthcare systems. For example, the respondents were asked to answer the question, “What do you think is the most pressing social problem in our country?” for 14 social problems; the value of the answer was “Yes” or “No”, which we think may simplify the different extent to which respondents are concerned about the accessibility and affordability of the healthcare system. Thus, we call for more nuanced measurements for perceived accessibility and affordability in the future. Moreover, although this study has revealed a representative understanding of the factors that influence perceived accessibility and affordability among older adults, this study was not able to reveal an in-depth understanding of the different health beliefs and health-seeking behaviors at the micro-level and the mechanisms behind such differences. Future research is encouraged to conduct a qualitative study (such as a case study) to unpack the complex phenomenon. Finally, since causal analysis is beyond the scope of our research, we did not incorporate the analysis on the mediating effects of self-rated SES on the relationship between perceived fairness and perceived accessibility and affordability.

## Figures and Tables

**Figure 1 healthcare-11-01818-f001:**
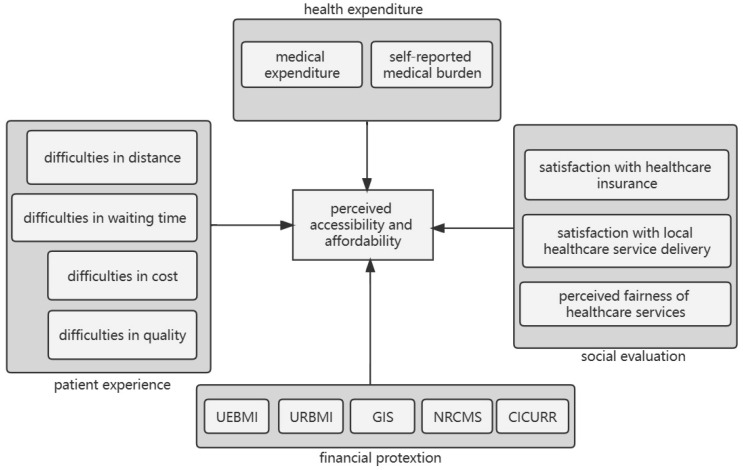
Conceptual framework.

**Table 1 healthcare-11-01818-t001:** Descriptive statistics of the variables of interest.

	Variable	Variable Type	Percentage	Mean (SD)	N
**Dependent variable**					
	Affordability and Accessibility	No = 0	45.5%	-	2169
		Yes = 1	54.5%		
**Demographic characteristics**					2169
	Gender	Male (1)	48.0%		
		Female (2)	52.0%		
	Age		-	64.92 (2.75)	
	Education level	Primary education or below (1)	50.3%		
		Secondary school (2)	26.9%		
		College or above (3)	17.8%		
	Annual income	-	-	18,569.12 (23,629.29)	
	Residency	Urban (1)	48.1%		
		Rural (2)	51.9%		
**Independent variables**					
Healthcare expenditure					
	Medical expenditure	Continuous	-	10,271.27 (33,948.3)	1986
	Medical burden	No = 0	58.6%	-	2169
		Yes = 1	41.4%		
Patient experience					
	Difficulties in distance	Continuous	-	3.17 (0.987)	1636
	Difficulties in waiting time	Continuous	-	3.14 (0.994)	1637
	Difficulties in cost	Continuous	-	2.37 (1.101)	1582
	Difficulties in quality	Continuous	-	3.18 (0.944)	1515
Social evaluation					
	Satisfaction with healthcare insurance	Continuous	-	6.23 (3.351)	2021
	Satisfaction with local healthcare service delivery	Continuous	-	2.80 (1.267)	2013
	Perceived fairness of healthcare services	Continuous	-	2.55 (1.420)	1957
Financial protection					
	Insurance coverage	No = 0	34.8%	-	2130
		Yes = 1	64.2%		
	Own UEBMI	No = 0	74.4%		1388
		Yes = 1	25.6%		
	Own URBMI	No = 0	88.0%		1388
		Yes = 1	12.0%		
	Own GIS	No = 0	92.4%		1388
		Yes = 1	7.6%		
	Own NRCMS	No = 0	40.0%		1388
		Yes = 1	60.0%		
	Own CICURR	No = 0	93.2%		1388
		Yes = 1	6.8%		

Note: Urban Employee Basic Medical Insurance (UEBMI), Urban Resident Basic Medical Insurance (URBMI), government insurance scheme (GIS), New Rural Cooperative Medical Scheme (NRCMS), and critical illness insurance for urban and rural residents (CICURR).

**Table 2 healthcare-11-01818-t002:** Descriptive results of the binary logistic regression models.

	Model 1			Model 2			Model 3			Model 4			Model 5		
	B (OR)	SE	95%CI	B (OR)	SE	95%CI	B (OR)	SE	95%CI	B (OR)	SE	95%CI	B (OR)	SE	95%CI
**Demographic characteristics**															
Gender	0.15 (1.16)	0.09	0.97–1.40	0.07 (1.08)	0.10	0.89–1.31	0.14 (1.15)	0.13	0.90–1.48	0.25 (1.29)	0.16	0.94–1.77	0.23 (1.26)	0.18	0.89–1.77
Age	−0.03 (0.97)	0.02	0.94–1.00	−0.03 (0.97)	0.02	0.94–1.00	−0.02 (0.98)	0.02	0.94–1.03	−0.03 (0.97)	0.03	0.92–1.03	−0.02 (0.98)	0.03	0.92–1.04
Education level															
Secondary school	0.26 * (1.30)	0.11	1.05–1.61	0.29 * (1.34)	0.12	1.06–1.69	0.23 (1.25)	0.15	0.94–1.68	0.36 (1.44)	0.19	0.99–2.11	0.17 (1.19)	0.21	0.78–1.79
College or above	0.39 ** (1.48)	0.13	1.14–1.91	0.40 ** (1.50)	0.14	1.14–1.97	0.31 (1.36)	0.18	0.96–1.93	0.40 (1.50)	0.24	0.95–2.37	0.26 (1.30)	0.25	0.80–2.12
Annual income	<0.01 (1.00)	<0.01	1.00–1.00	<0.01 (1.00)	<0.01	1.00–1.00	<0.01 (1.00)	<.01	1.00–1.00	<0.01 (1.00)	<0.01	1.00–1.00	<0.01 (1.00)	<0.01	1.00–1.00
Region	0.01 (1.01)	0.10	0.84–1.22	−0.04 (0.97)	0.10	0.79–1.18	0.07 (1.07)	0.13	0.83–1.39	−0.11 (0.89)	0.19	0.62–1.29	−0.04 (0.96)	0.20	0.65–1.43
**Healthcare expenditure**															
Medical expenditure				<0.01 (1.00)	<0.01	1.00–1.00	<0.01 * (1.00)	<.01	1.00–1.00	<0.01 (1.00)	<0.01	1.00–1.00	<0.01 (1.00)	<0.01	1.00–1.00
Medical burden				0.82 *** (2.28)	0.10	1.87–2.77	0.48 *** (1.61)	0.13	1.26–2.07	0.38 * (1.46)	0.16	1.06–2.02	0.24 * (1.27)	0.18	0.89–1.80
**Patient experience**															
Distance							−0.01 (0.99)	0.07	0.86–1.14	−0.04 (0.96)	0.09	0.81–1.15	−0.06 (0.95)	0.10	0.78–1.15
Waiting time							−0.05 (0.95)	0.08	0.82–1.11	0.09 (1.09)	0.09	0.91–1.30	0.13 (1.14)	0.10	0.93–1.39
Cost							−0.52 *** (0.60)	0.07	0.52–0.68	−0.64 *** (0.53)	0.09	0.45–0.63	−0.60 *** (0.55)	0.10	0.46–0.67
Quality							0.02 (1.02)	0.07	0.89–1.17	0.08(1.09)	0.09	0.91–1.30	0.13 (1.14)	0.10	0.93–1.39
**Financial protection**															
Own UEBMI										0.09 (1.10)	0.32	0.59–2.04	0.38 (1.46)	0.35	0.74–2.88
Own URBMI										−0.09 (0.92)	0.30	0.51–1.66	0.15 (1.16)	0.33	0.61–2.22
Own GIS										−0.19 (0.83)	0.36	0.41–1.68	−0.05 (0.95)	0.40	0.43–2.07
Own NRCMS										0.21 (1.24)	0.32	0.66–2.31	0.48 (1.61)	0.35	0.81–3.17
Own CICURR										−0.18 (0.84)	0.29	0.48–1.46	−0.11 (0.90)	0.31	0.49–1.65
**Social evaluation**															
Satisfaction with healthcare insurance													−0.04 (0.96)	0.04	0.90–1.03
Satisfaction with local healthcare delivery													−0.11 (0.89)	0.13	0.69–1.16
Fairness of healthcare services													−0.37 ** (0.69)	0.12	0.54–0.88
N	2060			1882			1283			817			727		
Nagelkerke R^2^	0.01			0.06			0.15			0.17			0.22		

Note: * *p* < 0.05, ** *p* < 0.01, *** *p* < 0.001; Beta (B), standard error (SE), odd ratios (exp(B), OR), and the associated 95% of confidence interval (CI); Urban Employee Basic Medical Insurance (UEBMI), Urban Resident Basic Medical Insurance (URBMI), government insurance scheme (GIS), New Rural Cooperative Medical Scheme (NRCMS), and critical illness insurance for urban and rural residents (CICURR).

## Data Availability

This study was authorized to use a publicly archived dataset from the Chinese Social Survey 2021 (link: http://csqr.cass.cn/index.jsp (accessed on 16 March 2023)).

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
