# Peer review of "Exploring Older Adults’ Perceived Affordability and Accessibility of the Healthcare System: Empirical Evidence from the Chinese Social Survey 2021"

_healthcare, 2023, doi:10.3390/healthcare11131818_

Round 1

Reviewer 1 Report

While the investigators attempted to genuinely address factors/determinates associated with perceived affordability and accessibility of the healthcare system in China, the variables included to assess the dependent variables appear to be inadequate. Common confounding variables including age, gender, income levels, educational background, location of participants (e.g., urban/rural), occupation, etc. were not included in the analysis.

The definition of the dependent variable may also be prone to bias. It was noted that participants were asked to answer the question: “What do you think is the most pressing social problem in our country?” The dependent variable is constructed as a binary variable, assigning a value of 1 if the elderly selected Kan Bing Nan, Kan Bing Gui as China’s most pressing social problem, and a value of 0 if not. What were the other options included under category ‘0’ and how do investigators ensure that the responses included in this category are not related to affordability and accessibility issues? This study might be well suited for a qualitative study where a detailed analysis of beliefs and perceptions (barriers, benefits) are examined using one of the public health theories (e.g., the health belief model or the theory of planned behavior). 

Need minor editing. Otherwise, the paper is well written. 

Reviewer 2 Report

This study addressed the challenges faced by the Chinese government in providing accessible and affordable healthcare services to its elderly population. The authors draw on data from the Chinese Social Survey 2021 to explore both objective and subjective factors affecting healthcare accessibility and affordability, including healthcare expenditure, patient experience, financial protection, and healthcare-related beliefs. Their findings suggest that self-reported medical burden, difficulty in the cost of healthcare services, and perceived fairness of public healthcare services were significantly associated with the older adults’ perceived affordability and accessibility in the healthcare system. There are some issues that could be addressed to strengthen the contributions of this paper.

The authors provide a nice review of the police background and the healthcare system reform in China. This is a plus for this research. While the paper explores both objective and subjective factors that contribute to public perceptions of healthcare accessibility and affordability among older adults in China. However, they do not offer a definition of what they mean by objective or subjective factors. They hint in the front end that policy factors such as reimbursement rates may act as the objective factor but they don’t measure those in their study.  This may make it difficult for readers to fully link their concepts to the measurements they utilized in this study.

I also suggest the authors lay out a conceptual framework. While the authors explore various independent variables that contribute to public perceptions of healthcare accessibility and affordability, there is no clear conceptual framework that guides their selection or organization of these factors. This may make it difficult for readers to fully understand how these variables relate to one another or how they contribute to overall public perceptions.

Regarding the dependent variable, it may be worth reconsidering whether categorizing the outcome as simple as yes or no is a suitable measure for evaluating accessibility and affordability. The survey question asks about “the most significant social problem in our country”, and just because individuals did not choose "Kan Bing Nan, Kan Bing Gui" does not mean they do not value accessibility and affordability as crucial factors. The comparison is between various social issues that may not necessarily fall under the medical service domain, assuming those who select “no” as no such concern are too brutal.

One thing that concerns me is your belief variables, specifically your satisfaction with healthcare insurance, healthcare service delivery, and fairness. These factors could simply reflect your perceptions of healthcare accessibility and affordability, essentially using the concept to predict itself. I strongly recommend conducting a correlation analysis to determine the degree of correlation between these factors before moving on to a regression analysis.

When conducting regression analysis, it is often recommended to consider various individual demographic, socioeconomic, and other relevant factors. For example, perceived fairness may be highly correlated with one’s SES status. The significance between perceived fairness and the outcome may just be due to the effect of SES.  It can also be beneficial to incorporate variables that account for disparities between rural and urban areas or regions in your model.

I am also curious if there is a possibility of reverse causality/endogeneity. Although this study does not aim to identify causal effects, it is important to consider how your predictors may affect the outcome. It is possible that those who have limited access to medical services may report a greater sense of unfairness. To ensure the reliability of your findings, you may want to conduct several robustness checks, such as restricting your sample to only high SES or formal employees who have greater access to medical services. This will help determine if your findings remain consistent.

Some limitations you may consider. It's important to note that cross-sectional survey data may not be able to fully account for all the confounding effects at play. Additionally, measuring accessibility and affordability with a single general item may be too crude to maintain internal and external validity.

Minor:

It is recommended to refrain from using terms such as "the aged," "elders," or "the elderly." It is preferable to use neutral language such as "older adults" to ensure inclusivity.

English language fine. Minor editing of English language required

Round 2

Reviewer 1 Report

It appears that the authors addressed the issues related to the inclusion of additional variables and provided clarifications on the outcome variable. 

It appears that the authors addressed the issues related to the inclusion of additional variables and provided clarifications on the outcome variable. 

Author Response

Response to Reviewer #1:

Comment #1

It appears that the authors addressed the issues related to the inclusion of additional variables and provided clarifications on the outcome variable.

Response:

Thanks for the comments, and we greatly appreciate the valuable time and effort made by the reviewer.

Comments on the Quality of English Language

It appears that the authors addressed the issues related to the inclusion of additional variables and provided clarifications on the outcome variable.

Response:

Thanks for the comments, and we greatly appreciate the valuable time and effort made by the reviewer.

Reviewer 2 Report

The revised version of this paper has made progress in addressing some of the concerns raised in the previous review round. However, there are still some remaining issues that could be further addressed.

Firstly, it is recommended that the conceptual model clearly illustrates the pathways from the predictors to the outcomes, rather than being solely a plot summary of the predictors.

When including income in the regression model, it is important to consider the potential skewness in its distribution. A log transformation may be preferable in such instances. Additionally, there seems to be inconsistency in the sample size (n) across all variables, and an explanation of how missing values were handled should be provided.

I am confused with the coding of your outcome variable. The negative correlation coefficients suggest a negative relationship. Assuming that "1" indicates participants who choose "Kan Bing Nan, Kan Bing Gui," which implies difficulty in accessing medical resources, the negative association with all social evaluation variables seems to indicate that greater perceived difficulties in distance, for example, result in a lower likelihood of experiencing access issues. Please clarify this interpretation.

It is necessary to verify if the regression table aligns with the interpretation provided. Furthermore, the interpretation of the regression findings is unclear. For instance, the statement "every one unit increase in the older adults' perceived fairness of public healthcare services, the odds ratio of increasing the older adults' perceived affordability and accessibility decreased by 31%" requires clarification on whether this refers to an increase or decrease in the odds ratio.

Moderate editing of English language required
